# Developing a Recommendation Model for the Smart Factory System

**Chun-Yang Chang [1], Chun-Ai Tu [2],\* and Wei-Luen Huang [3]**

1  Department of Intelligent Commerce, National Kaohsiung University of Science and Technology, Kaohsiung 807618, Taiwan; cyc@nkust.edu.tw
2  Business Intelligence School, National Kaohsiung University of Science and Technology, Kaohsiung 807618, Taiwan
3  Kaohsiung Veterans General Hospital, Kaohsiung 807618, Taiwan; wlhuang@vghks.gov.tw
\*  Correspondence: 1103405108@nkust.edu.tw

**Abstract:** In Industry 4.0, the concept of a Smart Factory heralds a new phase in manufacturing; the Smart Factory System (SFS) will have a huge demand in Taiwan. However, the cost of constructing a factory system will be high, and the complexity processes and introduction time must be considered. Thus, it is important to figure out how to grasp the key success factors for Smart Factories to reduce difficulties in the process, deal with the occurrence of problems, and improve the success rate of constructing Smart Factories. This research constructs an SFS recommendation model to make up for past research deficiencies in terms of recommendation. It combines the methodology of the Engel–Kollat–Blackwell Model (EKB Model) and the Modified Delphi Method to derive SFS recommendation indicators. Through analyzing weights, the ELECTRE II was used to obtain the importance of each dimension by calculating the Modified Compound Advantage Matrix. For prototype indicators, it reviewed the past literature to find out deficiencies and examined the world's four largest manufactories or computer technology corporations to analyze their Smart Factory solutions regarding the SFS function characteristics. The survey ran for several rounds with a group of five experts to amend indicators until a consensus was obtained. It proposed 64 indicators of 8 primary dimensions in total, based on the Updated Information System Success Model, and then added the concepts of SFS Function characteristics, Information Security, Perceived Value, Perceived Risk, and UI Design. According to the indicators, the framework and prototype of this system will provide solutions and references for purchasing SFS, the functions of which include SFS purchase ability analysis, demand analysis of manufacture problems, and raking and scoring of recommendation indicators. It will provide real-time ranking and the best alternative recommendations to suppliers, and will not only be referred to for design and modification but also enable the requirements to be closer to the users' demands.

**Keywords:** Smart Factory; recommendation model; recommendation system; Modified Delphi Method; ELECTRE II Method

## 1. Introduction

The Smart Factory has become quite important in the manufacturing field in recent years, as big data, cloud computing, and the Internet of Things (IoTs) have changed the way of manufacture, leading to Industry 4.0 [1–5]. The market value of Smart Factory was at 120.98 billion USD in 2016 and is expected to grow approximately 9.3% to 188.72 billion USD between 2017 and 2022. (Markets and Markets, 2017). Grand View Research estimated that the global smart manufacturing market size will reach 395.2 billion USD by 2025 (Grand View Research, 2017). From the above, it is evident that the Smart Factory is quite important. The Smart Factory System (SFS) is a solution that combines big data, cloud computing, and IoTs to drive manufacture automation, leading to better production efficiency [6–9]. In practical application, SFS is clouded, synchronizing data in real-time. Combined with

IoTs, it allows for transmitting information interactively by sensors, enabling real-time monitoring, better quality control, and data synchronization, which improves efficiency and reduces costs [10,11]. Big data analysis can predict future demand to find out potential customers, and interconnected procedure interactions assist in the decision-making of smart analysis [12].

There are more investments in SFS of enterprises and many suppliers have provided solutions for the Smart Factory. However, the Smart Factory field contains lots of complicated technologies and skills, such as cloud computing, IoTs, Smart Systems, and so on. Besides, most purchasers do not know much about these skills or abilities to select the appropriate solutions. As SFS is quite expensive, selecting a wrong SFS will lead to tremendous financial losses in the long-term; consequently, knowing how to select an appropriate SFS is imperative.

This study reviewed the traditional factory Information System research and realized that Information System purchasers mostly focused on quality, functionality, cost, and risk. In addition, the system only proposed SFS selection in terms of quality, functionality, cost, risk, and information security dimensions [13]. Most of the past research studies lack complete derivation processes in selection procedures, as dimensions are heterogeneous, and some focus only on certain production process. The core concept of the Smart Factory is to use information technology and services of the Internet of Things to integrate processes, so that production can run more professionally and operate efficiently [14]. At present, most of the research works focus on intelligent technology and equipment, and few discuss the construction goals of Smart Factory systems. Smart Factory system will have a huge demand in the market. However, the cost of constructing a factory system will be high, and the complexity of the processes and the introduction time must be considered. If the wrong Smart Factory model is selected, the company will waste a lot of money and time. Thus, solving this problem is the motivation of this research.

Traditional factories have some problems, such as the lack of systematic integration of plant sites, workshops, equipment, and personnel [15]. In addition, there is the issue of how to collect, store, and generate real-time statistics on the huge amount of data of the equipment, which may affect the decision-making efficiency [16]. Moreover, traditional manufacturing may cause environmental damage and consume a large amount of non-renewable energy, and the labor force is constantly shrinking. Compared with Smart Factories, IoT can be used to integrate business processes, and the production can be run in a more streamlined and efficient manner to bring about the benefits of high quality and low cost [17]. However, a traditional factory usually has already caused time and money losses when machine failure is found. This problem is easy to avoid for Smart Factories. For example, big data can help the machine to identify the cause of the failure in real time, and fully analyze the data to bring marketing benefits [16]. Therefore, Smart Factories can improve the problems of traditional factories; however, when it comes to constructing the Smart Factory, there are still many difficulties and challenges in the core technologies such as big data, artificial intelligence, IoT, cloud computing, etc. Facing these high-cost and innovative technologies, traditional factories often do not know how to construct Smart Factories, leading to a high failure rate [18]. Therefore, it is important to determine how to grasp the key success factors for Smart Factories to reduce difficulties in the process and tackle problems that arise in order to improve the success rate of constructing these systems. In the past, there were very few studies on the key success factors for the construction of Smart Factories, and they were only seen by scholars who compiled related studies on Industry 4.0 and sorted out 16 key success factors [19].

Therefore, based on the fact that most of Taiwan's traditional industries will be transformed into Smart Factories and few papers mention a successful construction model, the purpose of the research is to build a recommendation system of a Smart Factory suitable for the development environment of Taiwan's enterprises. Most of the past research lack complete derivation processes in selection procedures, as dimensions are heterogeneous, and some focus only on certain production process. Therefore, the purpose of this study is

to develop a Smart Factory Recommendation System (SFRS) to assist enterprises to select an appropriate SFS by SFS suppliers ranking, SFS suppliers scoring, and SFS purchase ability analysis. The system combined the Rapid Application Development Method and Modified Delphi Method to derive SFS recommendation indicators and proposed a four-stage system development methodology including the recommendation model construction, prototype indicators construction, amendment and simplification of the prototype indicator, and recommendations for system development. Considering the versatility and regionality of the SFS system, the design and selection of dimensions and indicators will target the industry environment in Taiwan and apply the Modified Delphi Method with the literature review and expert opinions found in this research.

Through the support system, the SFRS can make up for the lack of consideration factors generated by the information system indicators and the complicated interference factors of the operation. That is, the recommendation system is used to strengthen the assessment of quality, function, cost, risk, and information security in the process of building a Smart Factory. The development of a Smart Factory Recommendation System (SFRS) will assist enterprises to select an appropriate SFS by suppliers ranking, suppliers scoring, and purchase ability analysis.

## 2. Literature Review

### 2.1. Smart Factory

Traditional manufacture is no longer enough to satisfy large manufacturing demands, with many problems in existing in each step of the process. According past Smart Factory characteristics, it was found that all these characteristics are dispersed and inconsistent. There are some characteristics of the Smart Factory, such RFID scanner technology, that can be used for real-time production and monitoring to improve productivity [20]. RFID tag, Wi-Fi, Bluetooth, and ZIGBEE are technologies that can be used to make all Smart Factory devices interconnected with each other to drive automation [2]. In addition, the RFID can realize CPS through range detection and can be used for tracking items, production assistance, and process monitoring [21] and can use big data to analyze cloud information to improve production, operational efficiency, and monitoring [5,22].

Thus, this study defines the Smart Factory as "a highly digitized and interconnected production facility that uses sensors to gather information, and then analyze big data by cloud computing to improve manufacture efficiency and predict future demand".

### 2.2. Recommendation Systems

In order to solve the problem of big data, many studies have proposed related methods for recommendation systems. A recommendation system helps users obtain information that they seek, and a good recommendation system also can find out other relevant information for users. In general, recommendation systems can be divided into three types [23]: content-based, collaborative filtering, and hybrid methods, for reducing information overload. For content-based filtering, each item will be analyzed and ranked according to the score for a queried item, and a list of high-score items will be recommended for some queries. Meanwhile collaborative filtering is one of the most popular methods for recommendation systems [24]. The system calculates the analyzes the basis according to a user from a group that shares similar interests. It not only considers an item's contents but also the profiles of users and may discover the potential needs of users with two major approaches, user-based and item-based. Researchers further developed the hybrid method to predict the result and avoid the cold-start problem or new user problem [25]. For some specific circumstances, it is always hard to choose places to go from an endless number of options. Recommendation systems suggest items for users according to their explicit and implicit feedback information, such as ratings and reviews, which can help us make decisions that are more appropriate. The relevant studies and applications on recommendation systems are shown in Table 1.

**Table 1.** The applications on recommendation systems.

| Scholars | The Applications |
|---|---|
| Bogdan Walek, Vladimir Fojtik (2020) | A monolithic hybrid recommender system with a collaborative filtering used to recommend suitable movies according to the user's favorite and least favorite genres [26]. |
| Duygu Çelik Ertuğrul, Atilla Elçi (2019) | A personalized health recommender system is web-enabled and able to construct personalized health care with the key enabling technologies and major applications from successful case studies [27]. |
| Aysun Bozanta, Birgul Kutlu (2018) | A hybrid recommendation mode that integrates user-based and item-based collaborative filtering and content-based filtering together with contextual information to recommend new venues to users according to their preferences [28]. |
| Dong-Hui Yang, Xing Gao (2016) | The recommender systems help to coordinate the online supply chain with one retailer and two manufacturers to maximize profit by providing different choices and alleviating channel conflict [29]. |
| Selene Hernández-Rodríguez et al. (2016) | A recommender system based on a non-personalized approach and similar order circumstances integrates an indirect material recommender system to assist in warehouse tasks and to help new users create certain parts [30]. |
| Luis Del Vasto-Terrientes et al. (2015) | ELECTRE-TRI-B is proposed to handle assignments of alternatives on several levels of the hierarchy into a recommender system, focused on ordered classification with multiple conflicting criteria such as content, context, or cost, to find the most suitable alternatives [31]. |

Based on these articles, we found that few studies have been conducted to establish a recommendation system for Smart Factories. Therefore, this research is based on the functions of the recommendation system to determine the key the dimensions and prime influencing factors of Smart Factories, with the aim to recommend suitable models with indicators to factories, according to the current situation and conditions.

### 2.3. Updated Information System Success Model

The Information System Success Model was proposed by DeLone and McLean in 1992, to identify and explain the relationships between five crucial, representative dimensions: System Quality, Information Quality, Use, User Satisfaction, and Individual Impact, with the aim of resulting in Organization Impact, to evaluate the success of Information Systems. The model is shown in Figure 1 [32].

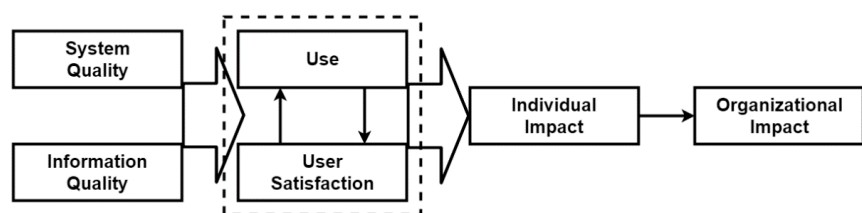

**Figure 1.** Information System Success Model [32].

System Quality and Information Quality will affect both Use and User Satisfaction, and these two dimensions will also influence each other interactively, and then result in Individual Impact, and finally lead to Organizational Impact. Later, in 2003, DeLone and McLean proposed the Updated Information System Success Model as Figure 2. We will compare the previous model and the current model below and highlight their differences.

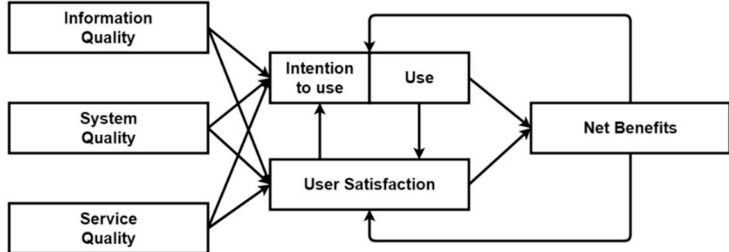

**Figure 2.** Updated Information System Success Model [32].

Since the advent of Updated D&M Information System Success Model, there has been a lot of information-system-associated research based on this model, measuring whether an Information System is successful by these six dimensions. Table 2 lists some of the related research using this model.

**Table 2.** Related research of the Information System Success Model.

| Scholars | Associated research |
|---|---|
| Guceglioglu & Demirors (2005) | Using Software Quality Characteristics to Measure Business Process Quality [33] |
| Chen et al. (2009) | Assessing the Quality of a Web-based Learning System for Nurses [34] |
| Rana et al. (2015) | Investigating the Success of an E-Government Initiative Validation of an Integrated Information System Success Model [12] |
| Nindiaswari et al. (2016) | Integration of Updated DeLone & McLean Success Model, KANO model and QFD to Analyze the Quality of an Information System [35] |
| Yang et al. (2017) | Understanding the Quality Factors That Influence the Continuance Intention of Students Toward Participation in MOOCs [36] |
| Al-Fadhli et al. (2018) | Understanding Health Professionals' Intention to Use Telehealth in Yemen Using the DeLone & McLean Information System Success Model [9] |

From the above, we can see that Updated Information System Success Model is a widely adopted approach to assess Information Systems. SFS has characteristics of the Information System; therefore, we use the Updated Information System Success Model as the basis of selection in this study, and then derive recommendation indicators through the three quality dimensions for SFS recommendation.

## 3. Methodology

According to the past research related to SFS, there is a paucity of research on SFS recommendations. Therefore, this study constructs an SFS recommendation model to make up for these research deficiencies. It combines the methodology of the Engel–Kollat–Blackwell Model (EKB Model) to ensure the needs of users and the Modified Delphi Method to derive SFS recommendation indicators from experts. Through analyzing weights, the ELECTRE II is used to obtain importance of each dimension by calculating the Modified Compound Advantage Matrix. The stages of operation are as follows.

Stage 1: Applying EKB Model for Prototype Recommendation Model Construction

According to the EKB Model, this includes (1) Demand Confirmation, (2) Supplier Information Collection, (3) Alternative Evaluation, and (4) Selection and Purchase, to construct the prototype recommendation model.

Stage 2: Prototype Indicators Construction by Updated Information System Success Model based on the EKB Model.

This study proposed 6 Information Quality, 5 System Quality, 5 Service Quality, 23 SFS Function characteristics, 16 Information Security, 8 Perceived Value, 6 Perceived Risk, and 5 UI Design recommendation indicators. In total, there are 74 recommendation indicators.

Stage 3: Amendment and Simplification of Prototype Indicator

After analyzing the suggestions from experts, we revised the recommendation indicators to 6 Information Quality, 5 System Quality, 4 Service Quality, 18 SFS Function characteristics, 13 Information Security, 7 Perceived Value, 6 Perceived Risk, and 5 UI Design. This led to a total of 64 recommendation indicators.

Stage 4: Recommendation System Development

This step involves providing SFS suppliers ranking, SFS suppliers scoring, and SFS purchase ability analysis, and assessing the system user satisfaction using the Updated Information System Success Model.

### 3.1. EKB Model

The Engel–Kollat–Blackwell Model (EKB Model) was developed in 1968 and revised in 1978. This model was originally used as a framework for organizing a great deal of customer behavior knowledge in order to enable enterprises to make better decisions [37]. This model argues that people receive information first and later store it in our experience, finally, we make an assessment with this information. This model is used to determine what affects customers' decisions and future consumption behaviors. There are four steps in the EKB Model:

Step 1: Demand Confirmation: Customers may have something they need; this will form a demand for customers to recognize their requirement.

Step 2: Information Collection: Once customers know their demands, they will begin searching for related information about certain products.

Step 3: Alternative Evaluation: When they found some products, the customers start to decide which alternatives are available (Assessment or Evaluation).

Step 4: Selection and Purchase: Customers will purchase the desired product according to the product evaluation results.

There is a study using the EKB model for decision-making process on the TAM model for "Attitude Toward Using" and "Behavioral Intentions to Use" and taking IoT as the object [38]. For EKB model, it provides a series of evaluation and decision-making processes on consumer behavior, including problem recognition, information search, program evaluation, purchase selection, and post-purchase behavior. Meanwhile, most econometric methods for studying choice start with the stages of evaluation of alternatives in the stylized EKB model. Taking the selection of a restaurant for leisure meals as an example, the stylized EKB model of the consumer decision process is used as a framework for developing different stages of the process and refers to problem recognition, information search, consideration set formation, evaluation of alternatives, and choice [39]. The EKB Model has been widely used and is still continuously used in understanding customer behaviors and supporting their decision-making processes. Through the literature review, the aim of this study is to understand a user's problems and needs first based on the EKB model, and then expand the empirical work of the factors in the recommendation system. In other words, we use the EKB Model to derive an SFS recommendation model with the aim of assisting an SFS purchaser when they are selecting SFSs.

### 3.2. Modified Delphi Method

Helmer and Dalkey first developed the Delphi Method in 1963. This is a structured, systematic, and communicative approach that relies on a panel of experts' subjective opinions and judgments to obtain objective recommendations or comments on a specific topic [40]. Later, in 1995, Murry and Hammons proposed the Modified Delphi Method. Figure 3 compares the Modified Delphi Method with the original Delphi Method. This modified version not only reduces the cost of designing and issuing these questionnaires

to the experts but also improves objectiveness because the prototype recommendation indicators are informed by a literature review rather than experts' discussions [41].

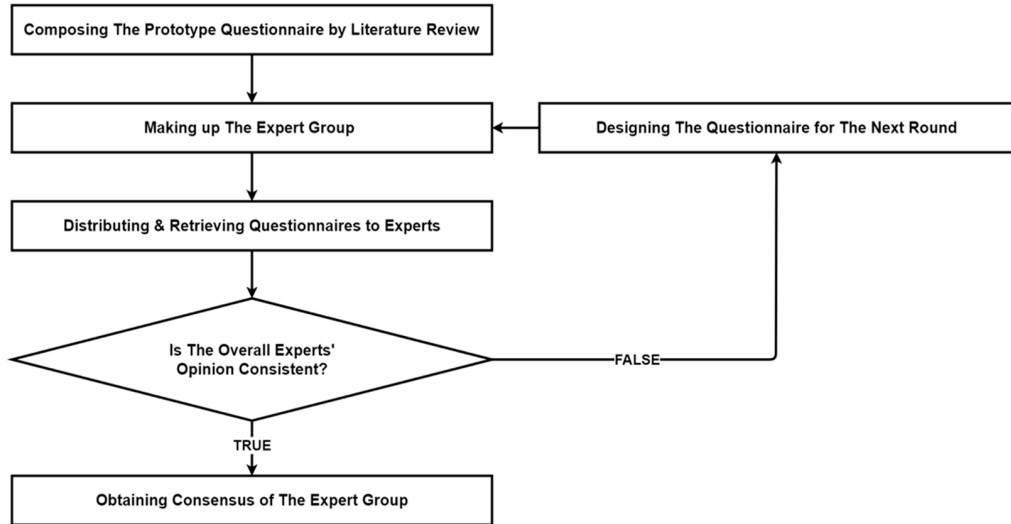

**Figure 3.** Modified Delphi Method steps [41].

From the above figure, it can be seen that the Modified Delphi Method is much more objective and timesaving than the original Delphi Method. Because of the advantages mentioned above, this study uses this method and finds five experts engaged in related jobs, related projects, and those researching or teaching related subjects of this domain. First, based on the literature review, to establish SFS prototype indicators, it runs the survey for several rounds with the expert group, amend indicators until a consensus is obtained.

### 3.3. ELECTRE II Method

ELECTRE I was first proposed by Benayoun in 1966. Its section version, ELECTRE II, was proposed by Roy in 1999. ELECTRE II calculates Concordance and Discordance indicators, and then sorts them by each advantage relation. ELECTRE II is widely used in decision making, as the limitations of ELECTRE I are greater than ELECTRE II, yielding fewer results [42]. The Modified Discordance was proposed by Sun in 2000, which yields better result when sorting and is able to select the best alternative [43].

The ELECTRE II Method is suitable for comparing between qualitative and quantitative data. This study has lots of factor considerations, such as Information Security, Perceived Value, Perceived Risk, and UI Design, but these data are hard to be quantitated. Therefore, this study will use ELECTRE II Method to obtain the importance of each dimension by calculating the Modified Compound Advantage Matrix.

## 4. Result of SFRS Development

### 4.1. Driving Recommendation Model

For the SFS recommendation model, the work is based on the four steps of the EKB Model and is shown in the selection processes as below.

Step 1: Customer Demand Confirmation Stage: Defining the requirements of the SFS for the enterprise.

Step 2: Supplier Information Collection Stage: Gathering information about SFS suppliers.

Step 3: Alternative Evaluation Stage: Based on recommendation dimensions that we organized the from literature review.

Step 4: Selection and Purchase Stage: Selecting an appropriate SFS from the recommendation results.

*4.2. Constructing Prototype Indicator*

This step of the research focused on reviewing the past literature to determine the deficiencies and gaps in the knowledge. Cost, Risk, Quality, and Functionality are the most considered dimensions of any Information System Selection. These form the foundation of the Updated Information System Success Model, which then adds concepts of SFS Function characteristics, Information Security, Perceived Value, Perceived Risk, and UI Design. Therefore, this study is organized based on DeLone and McLean's (2003) three quality types of Updated Information System Success Model, and proposes 6 Information Quality indicators, 5 System Quality indicators, and 5 Service Quality indicators. In total, there are 16 recommendation indicators [31], as shown in Table 3.

**Table 3.** SFS Recommendation indicator of three types of qualities.

| Dimension | SFS Recommendation Indicator |
|---|---|
| Information Quality | SFS can provide complete information. SFS can provide helpful information. SFS can provide up-to-date information. SFS can provide relevant information. SFS can provide understandable information. SFS can provide accurate information. |
| System Quality | SFS can improve original production procedures. SFS is reliable of its operation. SFS can yield information quickly. SFS can transmit information quickly. SFS is flexible to adapt changes of new functionalities. |
| Service Quality | SFS Suppliers are willing to assist customers actively. SFS Suppliers are reliable of their services. SFS Suppliers are trustworthy. SFS Suppliers can provide appropriate services. SFS Suppliers can provide practical assistance to solve the customer's problems. |

For adding the SFS function characteristics, this study took four of the world's largest manufactories or computer technology corporations as our SFS suppliers to analyze their Smart Factory solutions and then discuss their functionalities in depth. They include International Business Machines Corporation (IBM) as big data analysis and cloud computing, SIEMENS as digital factory design, General Electric Company (GE) as automation integration, and ASEA Brown Boveri (ABB) as robotic arm technology.

From the above solutions, this study organized SFS Function characteristics, and collated and proposed the following sub-dimensions: 1 Prediction Indicator, 1 Purchasing Indicator, 4 Production Indicators, 2 Quality Control Indicators, 2 Putting in Storage Indicators, 2 Inventory Control Indicators, and 11 Common Functionality Indicators. This led to a total of 23 recommendation indicators, as shown in Table 4.

**Table 4.** Sub-dimensions and indicators of SFS function characteristics.

| Sub-Dimension | Indicators |
|---|---|
| Prediction | SFS can automatically predict the relevant production demand information (e.g., time consumption, ingredients, cost, and expected sales revenue) based on past sales data. |
| Purchasing | SFS can automatically analyze the best ingredient suppliers based on their price and quality. |
| Production | SFS can automatically schedule. |
| | SFS can automatically control ingredients use. |
| | SFS can automatically contact the supplier to order ingredients when running short. |
| | SFS can automatically generate manufacturing progress reports. |
| Quality Control | SFS can automatically inspect and record the product's quality (including semi-finished and finished products) in real-time. |
| | SFS can automatically generate quality control reports. |
| Putting In Storage | SFS can automatically deliver finished products to the warehouse. |
| | SFS can perform specific processes depending on product properties (e.g., finished products need to be stored in a warehouse below 0 °C). |
| Inventory Control | SFS can keep up with real-time inventory situation. |
| | SFS automatically manages inventory to zero inventory requirements. |
| Common Functionality | SFS can instantly update all object information (e.g., purchase receipt, ingredients picking and warehousing information) by scanning (e.g., RFID, NFC and barcode) and synchronize data to the Cloud Management System. |
| | SFS can automatically generate reports (such as prediction reports, purchase reports, schedule reports, production progress reports, quality control reports, warehousing reports, maintenance record reports and abnormality record reports) |
| | SFS can immediately display production information and status on dashboards, screens and mobile devices. |
| | SFS can instantly display machine productivity and production load. |
| | SFS can automatically calculate the best combination of different operating machines to achieve optimal productivity, best quality, and lowest cost. |
| | SFS can record any machine events (e.g., abnormal event records, malfunction records, and regular maintenance records. |
| | SFS can automatically detect machine malfunction and ask for repair. |
| | SFS automatically issues periodic maintenance requests. |
| | SFS has abnormality self-troubleshooting mechanism. |
| | SFS can detect any source of danger and immediately issue an alert (e.g., fire, flood, and earthquake). |
| | SFS automatically analyzes the best business decision-making information. |

For adding Information Security, Perceived Value, Perceived Risk, and UI Design, this study organized concepts of CIA, Information Privacy, and Cybersecurity as Information Security. Through the literature review, 16 Information Security indicators were proposed. On Perceived Value, this study takes Expected Benefit, Perceived Quality, Brand Feeling, Sacrifice Value, Prestige Value, Emotional Value, Performance Value, and Investment Cost; a total of 8 Perceived Value recommendation indicators make up our SFS recommendation.

On Perceived Risk, this study takes Performance, Psychological, Social, Time, Physical, and Financial Risks; a total of 6 Perceived Risk recommendation indicators make up our SFS recommendation. On UI Design, it takes Efficiency, Memorability, Learnability, Satisfaction and Errors; there are a total of 5 UI Design recommendation indicators that make up our SFS recommendation.

### 4.3. Result of Prototype Indicators with Amendment and Simplification

In this stage, the research team visited five experts with backgrounds in manufacturing intelligent services and relation research, each with over 15 years of expertise in decision-making management. All of the experts work in different companies or institutions that are among the top 20 manufacturing companies in Taiwan, including two senior managers in the manufacturing department, a purchasing supervisor, a company leader, and a professor from a research institution.

This study analyzed the questionnaire through a five-point Likert scale (Likert 1932): "Very Important", "Important", "Neutral", "Unimportant", and "Very Unimportant" provided in our survey. Each item was checked with one of these five scales and suggestions or comments were left. In this way, the Reliability and Validity of the questionnaires were enhanced. This study performed two rounds of this survey and the experts' consensus was reached. The results of the survey are shown as below.

### 4.3.1. First Round of Modified Delphi Method

There is a total of 74 indicators of 8 dimensions in this round, including Updated Information System Success Model (16 indicators), SFS Function characteristics (23 indicators), Information Security (16 indicators), Perceived Value (8 indicators), Perceived Risk (6 indicators), and UI Design (5 indicators). It gave away 5 and retrieved all questionnaires. We found that only one item's SD value is greater than 1, which means this item does not obtain consensus, and 4 items' QD ranges between 0.61 and 1, which means these indicators reach only moderate consistency. After the First Round of the Modified Delphi Method, this study organized experts' suggestions (such as combined to a single item) and amended indicators according to their suggestions.

### 4.3.2. Second Round of Modified Delphi Method

There is a total of 64 indicators of 8 dimensions in this round. It gave away five and retrieved all questionnaires and found that all items' QD are less than 0.60, which means all indicators reach high consistency. After the Second Round of the Modified Delphi Method, we chose indicators that reach Mean and Mode value greater than 3.5 as our recommendation indicators. The 64 indicators of 8 dimensions in this round are shown in Figure 4.

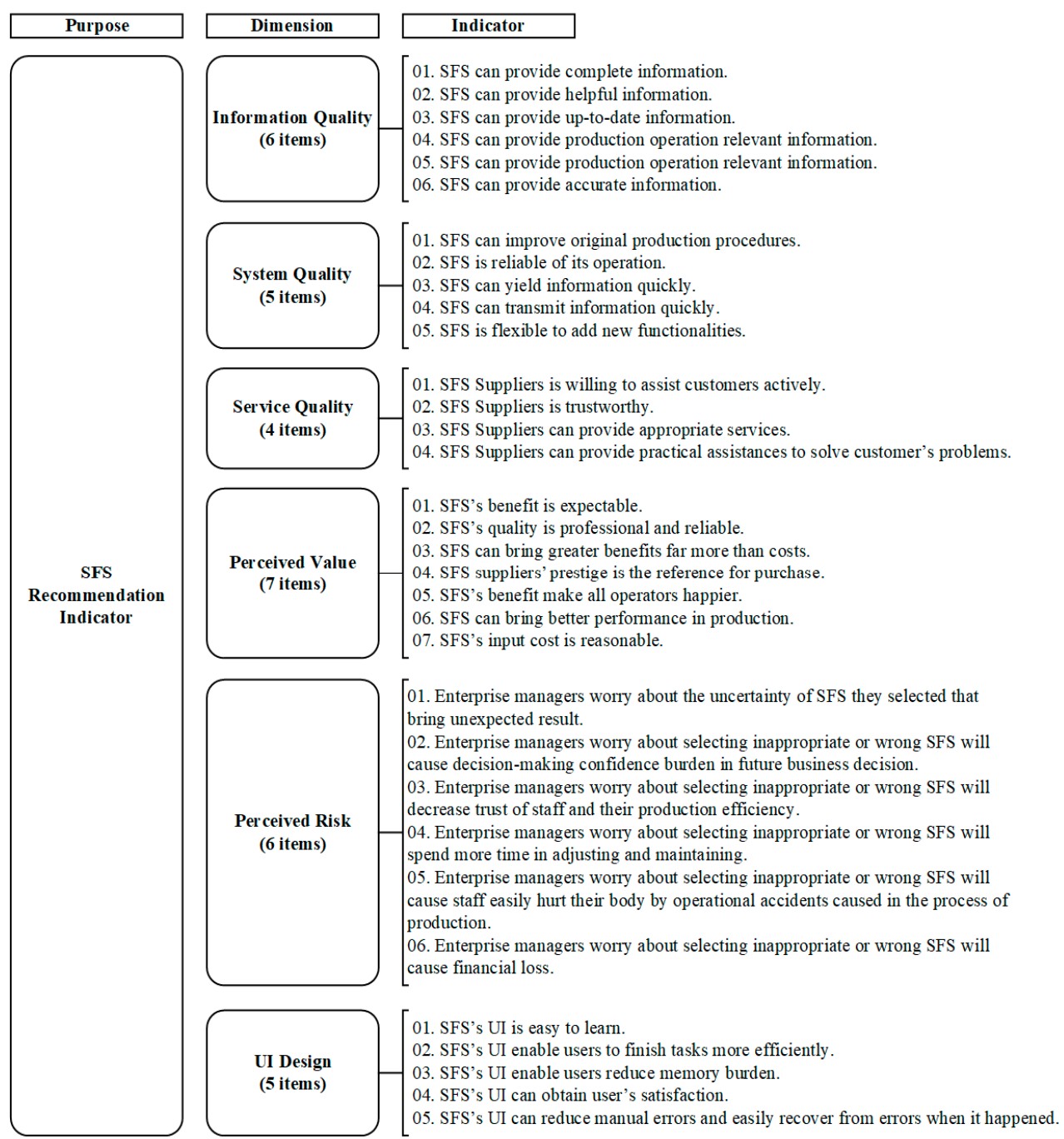

**Figure 4.** *Cont.*

| Purpose | Dimension | Indicator |
|---|---|---|
| **SFS Recommendation Indicator** | **SFS Function Characteristic (18 items)** | 01. SFS can automatically predict the relevant production demand information (e.g. time-consuming, ingredients, cost and expected sales revenue) based on past sales data.<br>02. SFS can automatically analyze the best ingredient suppliers based on their price and quality.<br>03. SFS can automatically schedule.<br>04. SFS can automatically control ingredients use.<br>05. SFS can automatically contact the supplier to order ingredients when running short.<br>06. SFS can automatically inspect and record the product's quality (including semi-finished and finished products) in real-time.<br>07. SFS can automatically deliver finished products to the warehouse and perform specific processes depending on products' properties (e.g. finished products need to be stored in a warehouse below 0 °C).<br>08. SFS automatically manages inventory to Zero Inventory requirements.<br>09. SFS can instantly update all object information (e.g. purchase receipt, ingredients picking, warehousing information) by scanning (e.g. RFID, NFC and barcode) and synchronize data to the cloud management system.<br>10. SFS can automatically generate reports (such as prediction reports, purchase reports, schedule reports, production progress reports, quality control reports, warehousing reports, maintenance record reports, abnormality record reports).<br>11. SFS can immediately display production information and status on dashboards, screens and mobile devices.<br>12. SFS can instantly display machine productivity and production load.<br>13. SFS can automatically calculate the best combination of different operating machines to achieve optimal productivity, best quality and lowest cost.<br>14. SFS can automatically detect machine malfunction and ask for repair.<br>15. SFS automatically issues periodic maintenance requests.<br>16. SFS has abnormality self-troubleshooting mechanism.<br>17. SFS can detect any source of danger and immediately issue an alert (e.g. fire, flood, earthquake).<br>18. SFS can automatically provide the best decision making information. |
| | **Information Security (13 items)** | 01. SFS ensures that all information is accessed as it is allowed.<br>02. SFS ensures that the data entered is secure.<br>03. SFS can ensure that the information in the Cloud System is correct.<br>04. SFS can ensure that the information in the cloud system is complete.<br>05. SFS can ensure that no data is leaked in the cloud when transmitting.<br>06. SFS can regularly back up data in different places.<br>07. SFS can automatically detect system vulnerabilities and propose countermeasures.<br>08. SFS can automatically monitor all manufacturing processes and issue alerts when suspicious events (such as wasting system energy) are detected.<br>09. SFS can immediately carry out protection measures when attacked or intruded.<br>10. SFS has Uninterrupted Power Systems (UPS) that prevent natural accidents.<br>11. SFS can automatically detect, record and analyze intrusions or attacks and alert immediately.<br>12. SFS can quickly recover after being compromised or attacked.<br>13. SFS can restore damaged data after being hacked or attacked. |

**Figure 4.** SFS recommendation indicators.

### 4.4. Result of Analyzing Alternatives Evaluation of SFS by Indicator Weight

#### 4.4.1. Factors Construction

This study divides factors into three levels. The first level is Objective: "SFS Alternatives Recommendation Indicators". The second level is Dimension, including "Information Quality", "System Quality", "Service Quality", "SFS Function characteristics", "Information Security", "Perceived Value", "Perceived Risk", and "UI Design"; there are eight dimensions in total. The third level consists of the Indicators, and there are 64 total items.

#### 4.4.2. Weights Construction

This study analyzes the weights of the following eight dimension and gives the most important dimension with the highest score, the second important dimension with the second highest score, and so on. The weights of each dimension in the SFS alternative recommendation are analyzed, as shown in Table 5 and the weights of recommendation indicators, such as "Information Quality" dimension, are shown in Table 6. The results of the sorting of all the indicators are shown in Table 7.

**Table 5.** The weight of dimensions for SFS.

| Dimension | Expert A | Expert B | Expert C | Expert D | Expert E | Total | Weight |
|---|---|---|---|---|---|---|---|
| Information Quality | 5 | 5 | 5 | 3 | 6 | 24 | 0.133 |
| System Quality | 4 | 8 | 3 | 5 | 5 | 25 | 0.140 |
| Service Quality | 8 | 2 | 8 | 6 | 7 | 31 | 0.172 |
| SFS Function Characteristic | 2 | 4 | 6 | 4 | 2 | 18 | 0.100 |
| Information Security | 7 | 7 | 7 | 7 | 8 | 36 | 0.200 |
| Perceived Value | 6 | 3 | 2 | 1 | 3 | 15 | 0.083 |
| Perceived Risk | 3 | 1 | 4 | 8 | 4 | 20 | 0.111 |
| UI Design | 1 | 6 | 1 | 2 | 1 | 11 | 0.061 |
| Total | 36 | 36 | 36 | 36 | 36 | 180 | 1 |

**Table 6.** The weight of SFS's Information Quality.

| Indicators Code | Expert A | Expert B | Expert C | Expert D | Expert E | Total | Weight |
|---|---|---|---|---|---|---|---|
| S-01 | 4 | 3 | 3 | 5 | 3 | 18 | 0.171 |
| S-02 | 5 | 6 | 5 | 6 | 4 | 26 | 0.248 |
| S-03 | 6 | 2 | 6 | 4 | 5 | 23 | 0.219 |
| S-04 | 2 | 1 | 4 | 3 | 6 | 16 | 0.152 |
| S-05 | 1 | 5 | 2 | 2 | 1 | 11 | 0.105 |
| S-06 | 3 | 4 | 1 | 1 | 2 | 11 | 0.105 |
| Total | 21 | 21 | 21 | 21 | 21 | 105 | 1 |

**Table 7.** The sorting of all the recommendation indicators.

| Dimension | Dimension Weight | Recommendation Indicator | Indicator Weight |
|---|---|---|---|
| Information Quality | 0.133 | SFS can provide helpful information. (S-02) | 0.248 |
| | | SFS can provide up-to-date information. (S-03) | 0.219 |
| | | SFS can provide complete information. (S-01) | 0.171 |
| | | SFS can provide production operation relevant information. (S-04) | 0.152 |
| | | SFS can provide understandable information. (S-05) | 0.105 |
| | | SFS can provide accurate information. (S-06) | 0.105 |

**Table 7.** *Cont.*

| Dimension | Dimension Weight | Recommendation Indicator | Indicator Weight |
|---|---|---|---|
| System Quality | 0.140 | SFS is reliable of its operation. | 0.280 |
| | | SFS can yield information quickly. | 0.267 |
| | | SFS can improve original production procedures. | 0.213 |
| | | SFS can transmit information quickly. | 0.160 |
| | | SFS is flexible to add new functionalities. | 0.080 |
| Service Quality | 0.172 | SFS Suppliers are trustworthy. | 0.380 |
| | | SFS Suppliers can provide practical assistance to solve the customer's problems. | 0.240 |
| | | SFS Suppliers are willing to assist customers actively. | 0.220 |
| | | SFS Suppliers can provide appropriate services. | 0.160 |
| SFS Function characteristics | 0.1 | SFS can automatically predict the relevant production demand information (e.g., time consumption, ingredients, cost, and expected sales revenue) based on past sales data. | 0.098 |
| | | SFS can automatically schedule. | 0.090 |
| | | SFS can automatically analyze the best ingredient suppliers based on their price and quality. | 0.065 |
| | | SFS can automatically control ingredients use. | 0.065 |
| | | SFS can automatically contact the supplier to order ingredients when running short. | 0.065 |
| | | SFS can immediately display production information and status on dashboards, screens and mobile devices. | 0.062 |
| | | SFS can detect any source of danger and immediately issue an alert (e.g., fire, flood, and earthquake). | 0.034 |
| | | SFS automatically issues periodic maintenance requests. | 0.033 |
| Information Security | 0.2 | SFS ensures entered data is secure. | 0.125 |
| | | SFS can regularly back up data in different places. | 0.116 |
| | | SFS ensures all information is accessed as it is allowed. | 0.110 |
| | | SFS can ensure no data is leaked in the Cloud when transmitting. | 0.097 |
| | | SFS can ensure information in the Cloud System is complete. | 0.092 |
| | | SFS can ensure information in the Cloud System is correct. | 0.090 |
| | | SFS can automatically detect system vulnerabilities and propose countermeasures. | 0.066 |
| | | SFS can automatically monitor all manufacturing processes and issue alerts when suspicious events (such as wasting system energy) are detected. | 0.066 |
| | | SFS has Uninterrupted Power Systems (UPS) that prevent natural accidents. | 0.064 |
| | | SFS can quickly recover after being compromised or attacked. | 0.055 |
| | | SFS can restore damaged data after being hacked or attacked. | 0.051 |
| | | SFS can automatically detect, record, and analyze intrusions or attacks and alert immediately. | 0.035 |
| | | SFS can immediately carry out protection measures when attacked or intruded. | 0.033 |

**Table 7.** *Cont.*

| Dimension | Dimension Weight | Recommendation Indicator | Indicator Weight |
|---|---|---|---|
| Perceived Value | 0.083 | SFS can bring greater benefits far more than costs. | 0.171 |
| | | SFS can bring better performance in production. | 0.157 |
| | | SFS's benefit is expectable. | 0.150 |
| | | SFS's quality is professional and reliable. | 0.143 |
| | | SFS's input cost is reasonable. | 0.143 |
| | | SFS's benefit make all operators happier. | 0.129 |
| | | SFS suppliers' prestige is the reference for purchase. | 0.107 |
| Perceived Risk | 0.111 | Enterprise managers worry that selecting inappropriate or wrong SFS will decrease the trust of staff and their production efficiency. | 0.219 |
| | | Enterprise managers worry about the uncertainty of the selected SFS that can bring unexpected results. | 0.200 |
| | | Enterprise managers worry that selecting inappropriate or wrong SFS will make them spend more time adjusting and maintaining. | 0.181 |
| | | Enterprise managers worry that selecting inappropriate or wrong SFS will cause decision-making confidence burden in future business decisions. | 0.171 |
| | | Enterprise managers worry about selecting inappropriate or wrong SFS that will cause financial loss. | 0.124 |
| | | Enterprise managers worry about selecting inappropriate or wrong SFS that will cause staff harm due to operational accidents in the process of production. | 0.105 |
| UI Design | 0.061 | SFS's UI enables users to finish tasks more efficiently. | 0.293 |
| | | SFS's UI is easy to learn. | 0.227 |
| | | SFS's UI can reduce manual errors and easily recover from errors when they occur | 0.200 |
| | | SFS's UI enables users reduce memory burden. | 0.160 |
| | | SFS's UI can obtain user satisfaction. | 0.120 |

According to the above research results, the following eight dimensions can be summarized.

(1) "Information Quality"

"SFS can provide helpful information" and "SFS can provide up-to-date information" are the top two most important items. As SFS gather information by RFID sensors and analyze business decision, useless information or out-of-date information might cause incorrect business decisions.

(2) "System Quality"

"SFS is reliable of its operation" and "SFS can yield information quickly" are the top two most important items. As SFS has different interconnected sub-systems among all six production procedures, if there are errors or problems that lead to downtime, they will result in extra human resources being used or higher costs to monitor production procedures. Therefore, the SFS's operation must be stable and yield information rapidly.

(3) "Service Quality"

"SFS Suppliers are trustworthy" and "SFS Suppliers can provide practical assistance to solve customer's problems" are the top two most important items. Since system users might sometimes encounter problems when manipulating the system, SFS suppliers should provide reliable solutions so that the system can be used appropriately. In summary, the

SFS supplier should be trustworthy; if SFS suppliers cannot solve customers' problem, this might lead to more problems, financial losses, or low production efficiency.

(4)    "SFS Function characteristics"

"SFS can automatically predict the relevant production demand information (e.g., time consumption, ingredients, cost, and expected sales revenue) based on past sales data" and "SFS can automatically schedule" are the top two most important items. As prediction and production schedule are cores of SFS, these are significantly different from the traditional factory. The prediction part uses big data analysis by analyzing past production and sales information, which can be used as references when proposing production strategies. The schedule part applies sensor technology and cloud management system, which can improve production productivity.

(5)    "Information Security"

"SFS ensures entered data is secure", "SFS can regularly back up data in different places" and "SFS ensures all information is accessed as it is allowed" are the top three most important items. SFS transmit, store, and analyze information by the cloud; although this allows for high efficiency in information sharing, more security issues must be considered, including the CIA, information privacy, and cybersecurity.

(6)    "Perceived Value"

"SFS can bring greater benefits far more than costs" and "SFS can bring better performance in production" are the top two most important items. SFS can automatically handle production procedures to improve efficiency and mitigate human forces, costs, and alleviate human errors. Although SFS is quite expensive, its automation brings plenty of benefits; therefore, the SFS price is not the only item we should consider, and it is less important than its benefits.

(7)    "Perceived Risk"

"Enterprise managers worry that selecting inappropriate or wrong SFS will decrease the trust of staff and their production efficiency" and "Enterprise managers worry about the uncertainty of the SFS they selected that can bring unexpected results" are the top two most important items. As SFS is a large-scale manufacture system, we must ensure that the system can be operated by operators properly and be sure of the results that it can yield.

(8)    "UI Design"

"SFS's UI enables users to finish tasks more efficiently" and "SFS's UI is easy to learn" are the top two most important items. An Information System should be as simple as possible and make operators accomplish their task more easily to reduce human errors.

4.4.3. ELECTRE II Method Evaluation Set Construction

According to evaluations from experts as "Very Important (VI)", "Important (I)", "Neutral (N)", "Unimportant (U)", and "Very Unimportant (VU)", this study uses the ELECTRE II Method to obtain the importance of each dimension by calculating the Modified Compound Advantage Matrix $E'$. The results are shown below as Table 8.

**Table 8.** Importance order with ELECTRE II method.

| Part S: | Information Quality | |
|---|---|---|
| | $E' = \begin{bmatrix} 0 & 1 & 1 & 1 & 1 \\ 0 & 0 & 1 & 1 & 1 \\ 0 & 0 & 0 & 1 & 1 \\ 0 & 0 & 1 & 0 & 1 \\ 0 & 0 & 1 & 1 & 0 \end{bmatrix}$ | Importance Order: VI > I > N = U = VU. |
| Part T: | System Quality | |
| | $E' = \begin{bmatrix} 0 & 1 & 1 & 1 & 1 \\ 0 & 0 & 1 & 1 & 1 \\ 0 & 0 & 0 & 1 & 1 \\ 0 & 0 & 1 & 0 & 1 \\ 0 & 0 & 1 & 1 & 0 \end{bmatrix}$ | Importance Order: VI > I > N = U = VU. |
| Part U: | Service Quality | Importance Order: VI > I > N = U = VU. |
| Part V: | SFS Function characteristics | Importance Order: I > VI > N > U = VU. |
| Part W: | Information Security | Importance Order: VI > I = N = U = VU. |
| Part X: | Perceived Value | Importance Order: I > VI > N = U = VU. |
| Part Y: | Perceived Risk | Importance Order: I > VI > N = U = VU. |
| Part Z: | UI Design | Importance Order: I > VI > N = U = VU. |

The experts believe that importance of each dimension is between Very Important (VI) and Important (I); therefore, the recommendation indicators proposed in this study have high importance.

In order to assist the SFS purchaser to select an appropriate SFS, this study uses the ELECTRE II Method to calculate the priority of available alternatives. Given a sample from the manufactory, the steps to obtain the result are shown below.

Step A. Setting Criteria Weight: Give each dimension a percentage weight: 13.3% to Information Quality, 14% to System Quality, 17.2 to Service Quality, 10% to SFS Function characteristics, 20% to Information Security, 8.3% to Perceived Value, 11.1% to Perceived Risk, and 6.1% to UI Design.

Step B. Using ELECTRE II Method: Evaluate the four suppliers by setting a preference rate for each dimension and use the ELECTRE II Method to calculate compound advantage relation among Company AA, BB, CC and DD, and finally yield Modified Compound Advantage Matrix $E'$ as shown in Table 9.

**Table 9.** Modified compound advantage matrix $E'$.

| | |
|---|---|
| $E' = \begin{bmatrix} 0 & 0 & 1 & 0 \\ 1 & 0 & 0 & 0 \\ 0 & 0 & 0 & 1 \\ 0 & 0 & 0 & 0 \end{bmatrix}$ | Result: BB > AA > CC > DD. |

Step C. Scoring the suppliers: The scores of the suppliers are BB (3.916), AA (3.418), CC (3.305), and DD (3.111), as shown in Table 10. Thus, this system recommends BB as the SFS supplier.

**Table 10.** Results of scoring suppliers.

| Dimension | Alternative | | | | Preference Rate (%) | Decimal Point |
|---|---|---|---|---|---|---|
| | **AA** | **BB** | **CC** | **DD** | | |
| Information Quality | 0.466 | 0.581 | 0.417 | 0.405 | 13% | 0.13 |
| System Quality | 0.326 | 0.418 | 0.394 | 0.313 | 11% | 0.11 |
| Service Quality | 0.537 | 0.574 | 0.513 | 0.389 | 15% | 0.15 |
| SFS Function characteristics | 0.349 | 0.404 | 0.330 | 0.328 | 10% | 0.1 |
| Information Security | 0.809 | 0.878 | 0.740 | 0.818 | 24% | 0.24 |
| Perceived Value | 0.300 | 0.325 | 0.280 | 0.288 | 9% | 0.09 |
| Perceived Risk | 0.420 | 0.477 | 0.422 | 0.375 | 12% | 0.12 |
| UI Design | 0.211 | 0.259 | 0.210 | 0.194 | 6% | 0.06 |
| Total | 3.418 | 3.916 | 3.305 | 3.311 | 100% | 1 |

### 4.5. Result of Recommendation System Development

This study applied a four-step customer decision process—EKB Model: Demand Confirmation, Information Collection, Alternative Evaluation and Selection and Purchase—to construct an SFRS, which is based on a PC and portable devices that go with Human–Machine Interfaces (HMI) to assist SFS purchasers and provide recommendations, shown in Figure 5. The input settings from the model and supplier database are selecting items, suppliers, scores from scales, and cost planning. The result of the output are weights of indicators, suppliers' ranking and scores, and purchase ability analysis.

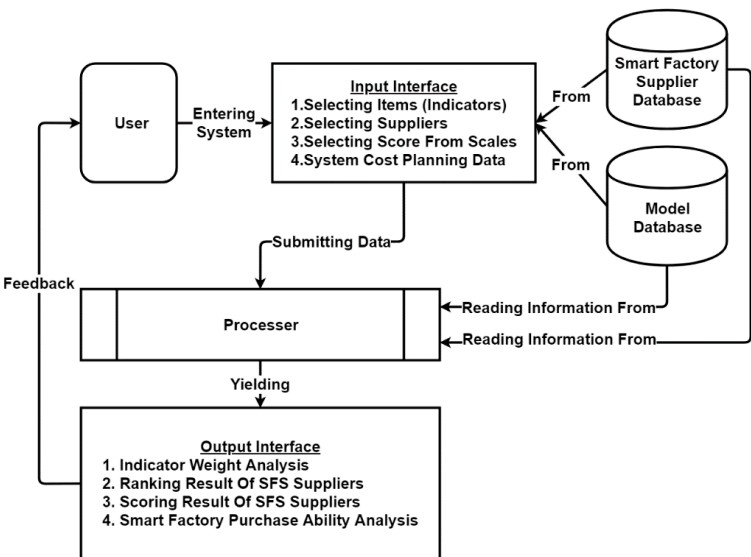

**Figure 5.** The Framework and prototype of the SFRS.

The framework and prototype of this system is based on content-based filtering recommendation to provide solutions and references on purchasing SFS, the functions of which are SFS purchase ability analysis; demand analysis of manufacture problems; raking and scoring of recommendation indicators. The development tools and specifications of this system include: (1) Operation system: Windows 10 64-bit Professional; (2) Integrated development environment: sublime text 3; (3) Programming language: HTML5, JavaScript, php; (4) Fore-End Framework: Bootstrap 4.0; (5) Database: MySQL; (6) Browser: google chrome version 67.0.3396.99 (64-bit); (7) Development kit: Eclipse 4.2.2, Android SDK r21.1, and UPnP kit (Cling Core 1.0.5). In addition, users can flexibly delete unnecessary indicators according to their demands and the system can automatically calculate recommendation results, providing real-time ranking and the best alternative recommendation. For suppliers, the results of this study can be used as references for the design and modification of the SFS and they also enable the requirements to be closer to the users' demands.

To understand the performance of the prototype recommendation system with User Satisfaction Assessment, four dimensions were assessed, including "Information Quality", "System Quality", "Service Quality", and "Business Performance". A total of 40 items were used to evaluate the SFRS's Use Satisfaction, and questionnaires were sent by e-mail to five supervisors of Company A. The results show that approximately 34% feel Highly Satisfied, 60% feel Satisfied, 6% feel Neutral, and no one feels Dissatisfied, Highly Dissatisfied. There were no answers that were Not Applicable.

In order to further understand the feasibility and suitability of the system, this study also selected three different manufacturing companies in Taiwan to verify the system according to different needs. The three manufacturing companies are Qingfa (screw manufacturing), Xiangda (plastic manufacturing), and Zhanjie (steel manufacturing). We personally reviewed the current status and development needs of the enterprises and listed the basic functions of smart manufacturing in accordance with each enterprise in detail under the recommendation system. At the same time, the supervisor of decision-making and the manufacturing department provided their opinions on Io, which will be used as the bases for the further development of the system.

## 5. Conclusions

This study adopts the Modified Delphi Method to identify the deficiencies in the past research, and then takes Updated Information System Success Model as the primary recommendation dimension; however, different systems have different requirements, so it is unable to use past selection methods for the SFRS. Therefore, in addition to taking Updated Information System Success Model as the base of recommendation indicators, the study also adds concepts, including Function characteristics of SFS; Information Security that protects enterprises' privacy and the CIA; Perceived Value for the evaluation of gain and sacrifice; Perceived Risk that enterprises have to take; and UI Design, which refers to whether a system is easy-to-use. These dimensions are extensions that make up for the deficiencies of the Updated Information System Success Model. The results of the system in this research can be used by enterprises; they provide solutions and references on purchasing SFS, whose functions are to provide purchase ability analysis of the SFS; to support demand analysis of manufacture problems; to rank and score recommendation indicators and flexibly delete unnecessary indicators by their demands; to automatically calculate recommendation results, which mitigates inconveniences of human calculation and provides real-time ranking and the best alternative recommendation. For SFS suppliers, the results of this study can be used as references for the design and modification of the SFS and they also enable the requirements to be closer to the users' demands.

Based on the results of the eight dimensions, the study suggests that purchasers should pay attention to: (1) information output accuracy, system stability, and information rapidness; (2) the supplier's proficiency which can explain system usage and solve possible problems; (3) demand accuracy and production schedule; (4) information security to mitigate data hack or leak; (5) how much added-value that a SFS can bring to the enterprises; (6) the SFS's expected results and operator system adaptability, and (7) ensuring that the UI design is easy to use. An SFRS developed for enterprises will provide solutions and references for purchasing SFS, with functions including purchase ability analysis, demand analysis of manufacture problems, and ranking and scoring of recommendation indicators.

The SFRS developed in this study can make the whole recommendation processes much quicker and smoother, eliminating complicated calculations and human errors. In the future work, the study will try to provide a new recommendation venue to users. That will be a hybrid recommendation model to integrate user-based and item-based models and then use collaborative filtering and content-based filtering recommendation systems for various domains.

**Author Contributions:** Conceptualisation, C.-Y.C.; data curation, C.-A.T.; formal analysis, C.-Y.C.; methodology, C.-A.T. and W.-L.H.; project administration, C.-Y.C.; sources, W.-L.H.; software, W.-L.H. and C.-Y.C.; supervision, C.-Y.C. All authors have read and agreed to the published version of the manuscript.

**Funding:** This research received no external funding.

**Institutional Review Board Statement:** Not applicable.

**Informed Consent Statement:** Not applicable.

**Data Availability Statement:** Not applicable.

**Conflicts of Interest:** The authors declare that they have no conflict of interest.

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
