# Peer review of "Developing a Recommendation Model for the Smart Factory System"

_applsci, doi:10.3390/app11188606_

Round 1

Reviewer 1 Report

The authors of the paper rightly argument that there is a paucity of research referring to Smart Factory System recommendation modeling. The authors combine Engel-Kollat-Blackwell Model and Modified Delphi Method to derive SFS Recommendation Indicators and propose a 4-Stage System Development Methodology.

The authors take Updated Information System Success Model as the base and add concepts of Smart Factory, Information Security, Perceived Value, Perceived Risk, and User Interface Design to establish Smart Factory System Prototype Indicators. The authors propose 64 Indicators of 8 primary Dimensions. The study results can assist enterprises in purchasing appropriate Smart Factory Systems and provide Smart Factory System Suppliers design references.

The paper is well built and informative.

I recommend to work on the Abstract to make it more suitable for a journal directed at readers generally acquainted with the research field. Please concentrate on the merits of the research (the additional information may then be provided in the developed Introduction and Literature Review sections).

In the Introduction please elaborate more on describing the research gap. Please describe the links between the research gap and the goal of the paper and research question. Please emphasize why the paper is important. Please add, perhaps in bullet points, what the main contributions of the paper to the field are.

While Section 2 Literature Review includes background on Smart Factory and Information System Success Model, an introduction to the field of Recommendation Systems (RS) is missing. The authors should elaborate on that, provide the definitions, and show that the RS field is widely studied nowadays and very important (in particular during the pandemic times, thanks to replacing the need for human recommendations). Please refer to some most recent papers in the field of RS, incl.  Applied Sciences journal paper titled Horizontal vs. vertical recommendation zones evaluation using behavior tracking,   LNDECT journal paper titled Evaluation of varying visual intensity and position of a recommendation in a recommending interface towards reducing habituation and improving sales,  Information journal paper titled A Comprehensive survey of knowledge graph-based recommender systems: technologies, development, and contributions,  Algorithms journal paper titled Towards cognitive recommender systems, etc.

Please elaborate on the Conclusions section. I would recommend to include a summary of the results as well as information on their limitations and on general future research directions.

There are some English language errors, but overall the paper is understandable. Please work on that. In particular please do not use contractions, but rather full forms (e.g. there's -> there is).

Good luck in improving the paper. I will be available to re-review it after amendments.

Author Response

The Comments from Reviewer #1

  1. I recommend to work on the Abstract to make it more suitable for a journal directed at readers generally acquainted with the research field. Please concentrate on the merits of the research (the additional information may then be provided in the developed Introduction and Literature Review sections).

We thank the reviewer for the very helpful comment. Regarding “Abstract”, we have revised the content of the abstract and focused on research methods, processes and results. Please refer to “Abstract: from the revised article.  

  1. In the Introduction please elaborate more on describing the research gap. Please describe the links between the research gap and the goal of the paper and research question. Please emphasize why the paper is important. Please add, perhaps in bullet points, what the main contributions of the paper to the field are.

We are grateful for the suggestion. To be more clear and in accordance with the reviewer concerns, we have added a brief description as follows. Please see Line 62-91 of the revised article.

“ The core concept of the smart factory is to use the information technology and services of the Internet of Things to integrate processes, …………. ………, the purpose of the research is to build a recommendation system of smart factory suitable for the development environment of Taiwan’s enterprises. Through the support system, it can make up for the lack of consideration factors generated by the information system indicators and the complicated interference factors of the operation. That is, the recommendation system is used to strengthen the assessment of quality, function, cost, risk, and information security in the process of building a smart factory. The develop of a Smart Factory Recommendation System (SFRS) will assist enterprises to select an appropriate SFS by suppliers ranking, suppliers scoring and purchase ability analysis.”

  1. While Section 2 Literature Review includes background on Smart Factory and Information System Success Model, an introduction to the field of Recommendation Systems (RS) is missing. The authors should elaborate on that, provide the definitions, and show that the RS field is widely studied nowadays and very important (in particular during the pandemic times, thanks to replacing the need for human recommendations). Please refer to some most recent papers in the field of RS, incl.  Applied Sciences journal paper titled Horizontal vs. vertical recommendation zones evaluation using behavior tracking,   LNDECT journal paper titled Evaluation of varying visual intensity and position of a recommendation in a recommending interface towards reducing habituation and improving sales,  Information journal paper titled A Comprehensive survey of knowledge graph-based recommender systems: technologies, development, and contributions,  Algorithms journal paper titled Towards cognitive recommender systems, etc.

Thank you for the above suggested. According to the suggestion, we add section 2.2 Recommendation systems and completely described the content and application of this theory. Besides, we organized the relevant literature in recent years into a table as Table 1 from the revised article.

  1. Please elaborate on the Conclusions section. I would recommend to include a summary of the results as well as information on their limitations and on general future research directions.

We thank for the suggestion. To be more clear, we have revised the conclusion and added the future works. Please see “Section 5. Conclusion” of the revised article.

  1. There are some English language errors, but overall the paper is understandable. Please work on that. In particular please do not use contractions, but rather full forms (e.g. there's -> there is).

Thanks very much for the suggestions. We have corrected the errors including not to use contractions as Line 151, 163, 175, 282, 437 from the revised article.   

 Thanks very much for taking your time to review this manuscript. We really appreciate all your comments and suggestions! Please find my revisions/corrections in the resubmitted files. 

Reviewer 2 Report

The are several issues with the document. The most relevant is that there is no clear connection between the proposed model and smart factories. How will the proposed method improve or use the resources that are available on smart factories in relation to traditional factories?

Another issue is the claim that this is a gap in the research, although there are a limited number of papers on the subject a search for "industrial IOT EKB model" return some positive results.

The claim on line 92 is too strong for the presented SOTA, how does the Information System Success Model relate to the EKB model?

Finally, at the end of the introduction, there is no mention of the structure of the document.

Author Response

The Comments from Reviewer #2

  1. The are several issues with the document. The most relevant is that there is no clear connection between the proposed model and smart factories. How will the proposed method improve or use the resources that are available on smart factories in relation to traditional factories?

We thank the reviewer for the very helpful comment. Regarding “the connection between the proposed model and smart factories”, we have revised the content and add some parts on section 1. Introduction, and focused on research methods, processes and results. Please see Line 50-91, 163-168, 243-253, from the revised article.

Line 50-91 : “There are more investments in SFS of enterprises and lots of suppliers provided solutions for Smart Factory. However, the Smart Factory field contains lots of complicated technologies and skills, such as Cloud Computing, IoTs, Smart Systems and so on. Be-sides, most purchasers don’t know much about these skills or abilities to select these solutions. As SFS is quite expensive, hence selecting a wrong SFS will lead tremendous financial losses in the long-term perspective; consequently, knowing how to select an appropriate SFS is imperative.

This study reviewed traditional factory Information System research and realized that Information System purchaser mostly focused on Quality, Functionality, ………., Smart factories can improve the problems of traditional factories, but how to con-struct a smart factory, there are still many difficulties and challenges in the core technologies such as big data, artificial intelligence, Internet of things, cloud computing, etc. Facing these high-cost and innovative technologies……..

Therefore, based on the fact that most of Taiwan’s traditional industries to be trans-formed into smart factories and few papers mention the successful construction model, the purpose of the research is to build a recommendation system of smart factory suitable for …………..…, suppliers scoring and purchase ability analysis.”

Line 163-168 : ” According to past related research of SFS, there is a paucity of research referring to SFS recommendation. Therefore, this study constructs SFS recommendation model to make up past research deficiencies on recommendation. It combines the methodology of Engel-Kollat -Blackwell Model (EKB Model) and Modified Delphi Method to derive SFS recommendation indicators. Through analyzing weights, the ELECTRE II is used to obtain importance of each dimension by calculating Modified Compound Advantage Matrix.”

Line 243-253: ”For adding SFS function characteristic, this study organized the world largest 4 manufactories or computer technology corporations as our SFS suppliers to analyze their Smart Factory solution, and then discussed their functionalities in depth. They are including International Business Machines Corporation (IBM) as Big Data Analysis and Cloud Computing, SIEMENS as Digital Factory Design, General Electric Company (GE) as Automation Integration, ASEA Brown Boveri (ABB) as Robotic Arm Technology.

From the above solutions, this study organized SFS Function Characteristic, and collated and proposed the following sub-dimensions: 1 Prediction Indicator, 1 Purchasing Indicator, 4 Production Indicators, 2 Quality Control Indicators, 2 Putting In Storage Indicators, 2 Inventory Control Indicators and 11 Common Functionality Indicators, totally 23 recommendation indicators, as shown in Table 4.”

  1. Another issue is the claim that this is a gap in the research, although there are a limited number of papers on the subject a search for "industrial IOT EKB model" return some positive results.

We are grateful for the suggestion. To be more clear and in accordance with the reviewer concerns, we have added a brief description as follows. Please see Line 170-172, 184-187, 62-70 of the revised article.

Line 170-172: “Engel-Kollat-Blackwell Model (EKB Model) in 1968 and revised this model to a better version in 1978. This model was originally used as a framework for organizing a great deal of customer behavior knowledge in order to enable enterprises to make better deci-sions[34].”

Line 184-187: “EKB Model has been widely used and still continuously used in understanding customer behaviors and supporting the decision-making process. Therefore, in this study, we use EKB Model to derive SFS recommendation model in order to assist SFS purchaser when they are selecting SFSs.”

Line 62-70: “The core concept of the smart factory is to use the information technology and services of the Internet of Things to integrate processes, so that production can run more profession-ally and operate efficiently [14]. At present, most of the researches focus on intelligent technology and equipment, and few discuss the construction goals of smart factory systems. Smart factory system will have a huge demand in the market. However, the cost of constructing a factory system will be high, and the complexity processes and the introduction time must be considered. If the wrong smart factory model is selected, the company will waste a lot of money and time, so how to solve this problem is also the motivation of this research.”

  1. The claim on line 92 is too strong for the presented SOTA, how does the Information System Success Model relate to the EKB model?

Thank you for the above suggested. According to the suggestion, we add section 4.5 to completely describe the content and application. Please see Line 135-139, 151-153, 157-161, 170-173, 184-187, 163-168, 414-420 of the revised article.

Line 135-139: “Information System Success Model was proposed by DeLone & McLean in 1992, by identifying and explaining the relation among 5 most crucial and representative dimensions: System Quality, Information Quality, Use, User Satisfaction, Individual Impact, and finally result in Organization Impact that evaluates the success of Information Systems.”

Line 151-153: “Since the advent of Updated D&M Information System Success Model, there are a lot of information-system-associated research based on this model - Measuring whether an Information System is successful by these 6 Dimensions.”

Line 157-161: “From the above, we can find that Updated Information System Success Model is a widely adopted approach to assess Information Systems. SFS has characteristics of the Information System; therefore, we use the Updated Information System Success Model as the base of selection in this study, and then we will derive recommendation indicators by 3 Quality Dimensions for SFS recommendation.”

Line 170-173: ” Engel-Kollat-Blackwell Model (EKB Model) in 1968 and revised this model to a better version in 1978. This model was originally used as a framework for organizing a great deal of customer behavior knowledge in order to enable enterprises to make better decisions[34].”

Line 184-187: ”EKB Model has been widely used and still continuously used in understanding customer behaviors and supporting the decision-making process. Therefore, in this study, we use EKB Model to derive SFS recommendation model in order to assist SFS purchaser when they are selecting SFSs.”

Line 163-168: ” According to past related research of SFS, there is a paucity of research referring to SFS recommendation. Therefore, this study constructs SFS recommendation model to make up past research deficiencies on recommendation. It combines the methodology of Engel-Kollat -Blackwell Model (EKB Model) and Modified Delphi Method to derive SFS recommendation indicators. Through analyzing weights, the ELECTRE II is used to obtain importance of each dimension by calculating Modified Compound Advantage Matrix.”

Line 414-420: “This study applied 4-Step Customer Decision Process - EKB Model: Demand Confirmation, Information Collection, Alternative Evaluation and Selection & Purchase to con-struct SFRS, which based on the PC and portable devices that go with Human-Machine Interfaces (HMI) to assist SFS purchaser for SFS recommendation, shown as Figure 5. The input settings from model and supplier database, which are selecting items, suppliers, scores from scales and cost planning. The result of the output are weights of indicators, suppliers’ ranking and scores and purchase ability analysis.”

  1. Finally, at the end of the introduction, there is no mention of the structure of the document.

Thanks very much for the suggestions. We have explained the structure more clearly. First, we described the background and needs of SFS, please see Line 50-53 and Line 87-61. And then added two parts to explain the progress of this article on Line 71-90.

Line 50-53: “There are more investments in SFS of enterprises and lots of suppliers provided solutions for Smart Factory. However, the Smart Factory field contains lots of complicated technologies and skills, such as Cloud Computing, IoTs, Smart Systems and so on. Besides, most purchasers don’t know much about these skills or abilities to select these solutions.”

Line 87-61: “This study reviewed traditional factory Information System research and realized that Information System purchaser mostly focused on Quality, Functionality, Cost and Risk. In addition, only it proposed SFS Selection by Quality, Functionality, Cost, Risk and Information Security dimensions [13]. Most of past research lack of complete derivation processes in selection procedures, as dimensions are heterogeneous, some focus on certain production process.”

Line 71-90: “Smart factories can improve the problems of traditional factories, but how to con-struct a smart factory, there are still many difficulties and challenges in the core technologies such as big data, artificial intelligence, Internet of things, cloud computing, etc. Facing these high-cost and innovative technologies, traditional factories often do not know how to construct smart factories and then lead to a high failure rate [15]. Therefore, how to grasp the key success factors for smart factories to reduce the difficulties in the process ………….,  and they were only seen by scholars who compiled related studies on Industry 4.0 and sorted out 16 key success fac-tors[16].

Therefore, based on the fact that most of Taiwan’s traditional industries to be trans-formed into smart factories and few papers mention the successful construction model, the purpose of the research is to build a recommendation system of smart factory suitable for the development environment of Taiwan’s enterprises. Through the support system, it can make up for the lack of consideration factors generated by the information system indicators and the complicated interference factors of the operation. That is, the recommendation system is used to strengthen the assessment of quality, function, cost, risk, and in-formation security in the process of building a smart factory. The develop of a Smart Fac-tory Recommendation System (SFRS) will assist enterprises to select an appropriate SFS by suppliers ranking, suppliers scoring and purchase ability analysis.”

Thanks very much for taking your time to review this manuscript. We really appreciate all your comments and suggestions! Please find my revisions/corrections in the resubmitted files. 

Reviewer 3 Report

Dear authors,

I was pleased to read your article. I also studied older versions of the article and your edits. The manuscript has significantly improved as compared to the previous version.

I recommend emphasizing the associated research question in the introduction section. Future research opportunities should also be better emphasized. It is possible to emphasize the boundaries of research and the conditions of use of the selected method. The conclusion could be extended about the possibility of a next specific application of the proposed solution in practice. Future research directions may also be highlighted. In the manuscript, please emphasize the novelty of your approach.

Overall the article is interesting.

My review is not a critique, it's just my recommendations and my view of the post.

I wish you good luck in further research.

Kind regards,

Reviewer

Author Response

Thank you very much for taking your time to review this manuscript. We are really grateful for the opportunity to revise the article according all comments and suggestions! Please find my revision in the resubmitted file. 

I recommend emphasizing the associated research question in the introduction section. Future research opportunities should also be better emphasized. It is possible to emphasize the boundaries of research and the conditions of use of the selected method. The conclusion could be extended about the possibility of a next specific application of the proposed solution in practice. Future research directions may also be highlighted. In the manuscript, please emphasize the novelty of your approach.

We thank the reviewer for the very helpful comment. Regarding it, we have revised the Introduction and Conclusion.

  1. For emphasizing the research question, we add some content to be clear, please see Line 89-102.
  2. About the boundaries of research, the conditions of use of the selected method and the novelty of the approach, we made some explanations, please see Line 103-109.
  3. For Future research opportunities and directions, we modified the content in the conclusion. Please see Line 552-557.

Line 89-102:

Therefore, how to grasp the key success factors for smart factories to reduce the difficulties in the process and the occurrence of problems, to improve the success rate of constructing smart factories are very important topics. In the past, there were very few studies on the key success factors for the construction of smart factories, and they were only seen by scholars who compiled related studies on Industry 4.0 and sorted out 16 key success fac-tors [19].

Therefore, based on the fact that most of Taiwan’s traditional industries to be trans-formed into smart factories and few papers mention the successful construction model, the purpose of the research is to build a recommendation system of smart factory suitable for the development environment of Taiwan’s enterprises. Most of past research lack of complete derivation processes in selection procedures, as dimensions are heterogeneous, some focus on certain production process. Therefore, the purpose of this study is to develop a Smart Factory Recommendation System (SFRS) to assist enterprises to select an appropriate SFS by SFS Suppliers Ranking, SFS Suppliers Scoring and SFS Purchase Ability Analysis.

Line 103-109:

It combined Rapid Application Development Method and Modified Delphi Method to derive SFS recommendation indicators and proposed 4-Stage system development methodology including recommendation model construction, prototype indicators construction, amendment & simplification of prototype indicator and recommendation system development. Considering the versatility and regionality of SFS system, the design and selection of dimensions and indicators will target industry’s environment in Taiwan and apply Modified Delphi Method with a large amount of literature review and expert opinions for the scope and limitation in this research.

Line 552-557:

SFRS developed from this study can make the whole recommendation processes much quicker and smoother, and eliminate complicated calculation and human errors. For the future woks, the study will provide a new recommend venues to users that a hybrid recommendation model is proposed to integrate user-based and item-based on collaborative filtering and content-based filtering recommend systems in order to concerning various domains.

Reviewer 4 Report

The paper is interesting, based on the analysis of available methods and models, which were used by the authors to develop a Recommendation Model for Smart Factory System. Obviously, the scope of the paper relates to the journal topic. However, important clarifications and corrections are still required.

How are the weight of dimensions for SFS and indicator weight calculated? Does the way they are calculated provide an estimate of the significance, importance and prominence of dimensions? Shouldn't methods from statistical analysis such as cronbach's alpha be used for this purpose?

How were the experts selected for the evaluations carried out?

In addition, a certain system architecture is presented in section 4.5. How was it implemented in a real environment? What technologies were used, can examples of its operation be presented?
Moreover, If users can flexible delete unnecessary indicators by their demands, can they add new ones? If they can add new ones how are the weights, which are based on expert judgement, calculated?

Is the assessment of the proposed solution made on the basis of 5 supervisors of Company A reliable? Was it only one company?

Conclusions are very general. There is no reference to the obtained results in a reliable way, i.e. what effects will be brought by the application of the presented recommendations in a real Smart Factory environment?

Author Response

Thank you very much for taking your time to review this manuscript. We are really grateful for the opportunity to revise the article according all comments and suggestions! Please find my revision in the resubmitted file. 

  1. How are the weight of dimensions for SFS and indicator weight calculated? Does the way they are calculated provide an estimate of the significance, importance and prominence of dimensions? Shouldn't methods from statistical analysis such as cronbach's alpha be used for this purpose?

We are grateful for the suggestion. To be more clear, we have added the table and provide relevant explanations. Please see Line 362-367 & 390.

The weights are mainly based on the values of the dimensions and indicators given by each expert, and the weights are determined through integration. First, we determine the weights of the general dimensions, and then evaluate the indicators of each dimension. We added an additional table 6 to illustrate the sorting of all the recommendation indicators as table 7.

In order to define the rationality of the dimensions and indicators, we use two rounds of Modified Delphi Method to find out the consensus from experts that are used for determine the weights. The main purpose of the Delphi method is to form a situation that enables experts to gain a consensus and obtain the final result. Thus, in this section, we obtain weights from expert experience to show the relative importance of factors in the overall evaluation, and then conduct a comprehensive evaluation of the indicators.

Line 362-367:

It analyzes the weights of the following 8 dimension and gives the most important dimension with the highest score, the second important dimension with a second highest score, and so on. This study analysis the weights of each dimension in SFS alternative recommendation, as shown in Table 5 and the weights of recommendation indicators, such as “Information Quality” dimension shown in Table 6 and so on. The analysis result of all the indicators sorting as shown in Table 7.

  1. How were the experts selected for the evaluations carried out?

We thank the reviewer for the very helpful comment. Regarding it, we have added some content to illustrate the source and background of the experts. Please see Line 325-330.

Line 325-330:

In this stage, the research visited 5 experts whose background in manufacturing intelligent services and relation research, and are with working experience for more than 15 years in decision-making management. All of the experts work in different companies or institutions that are among the top 20 manufacturing companies in Taiwan including two senior managers in the manufacturing department, a purchasing supervisor, a company leader, and a professor from a research institution.

  1. In addition, a certain system architecture is presented in section 4.5. How was it implemented in a real environment? What technologies were used, can examples of its operation be presented?

Moreover, If users can flexible delete unnecessary indicators by their demands, can they add new ones? If they can add new ones how are the weights, which are based on expert judgement, calculated?

Thank you for the above comment. We have listed some developing tools and specifications to illustrate how the system work. Please see Line 490-495.

In addition, the indicators in this research are based on the common needs and functions of the current smart manufacturing system, and have included the content of the literature and the factors recognized by the experts. It can be a reference for transforming smart manufacturing and reduce the risk of failure. Therefore, the current system can only consider its own conditions under the existing indicators, and cannot directly add new indicators. The suggestions of reviewer are important directions for future research, and we will continue to develop it in the follow-up direction.

Line 490-495:

Developing tools and specifications of this system includes (1) operation system: Windows 10 64-bit Professional; (2) integrated development environment: sublime text 3; (3) programming language: HTML5, JavaScript, php; (4) Fore-End Framework: Bootstrap 4.0; (5) database: MySQL; (6) browser: google chrome version 67.0.3396.99 (64-bit); (7) development kit: Eclipse 4.2.2, Android SDK r21.1 and UPnP kit (Cling Core 1.0.5).

  1. Is the assessment of the proposed solution made on the basis of 5 supervisors of Company A reliable? Was it only one company?

We are grateful for the good suggestion. In order to further understand the feasibility and suitability of the system, we took some days to selected three different manufacturing industries to verify the system for different needs in the verification stage. Please see Line 510-518.

Line 510-518:

In order to further understand the feasibility and suitability of the system, this study also selected three different manufacturing companies in Taiwan to verify the system according to different needs. The three manufacturing companies are Qingfa screw manufacturing, Xiangda plastic manufacturing and Zhanjie steel manufacturing. This research personally reviewed the current status and development needs of the enterprises, and listed the basic functions of smart manufacturing in accordance with the enterprise in detail under the recommendation system. At the same time, the supervisor of decision-making and the manufacturing department provided their opinions that will be used as the basis for further development of the system.

  1. Conclusions are very general. There is no reference to the obtained results in a reliable way, i.e. what effects will be brought by the application of the presented recommendations in a real Smart Factory environment?

Thank you for the above comment. We add some content to illustrate the research results and contributions, as well as the application functions that can be substantively operated in the enterprise. Please see Line 529-537.

Line 529-537:

The results of the system in this research are for enterprises, they provide solutions and references on purchasing SFS whose functions are (1) to provide purchase ability analysis of SFS; (2) to support demand analysis of manufacture problems; (3) raking and scoring of recommendation indicators and to flexibly delete unnecessary indicators by their de-mands; (4) to automatically calculate recommendation results, which mitigates inconveniences of human forces calculation and provides real-time ranking and the best alter-native recommendation. For SFS Suppliers, results of this study can be a reference on de-sign and modification of the SFS, enabling the SFS's requirements to be closer to users' demands.

Round 2

Reviewer 2 Report

Most of my previous concerns were not properly addressed:

1. How will the proposed method improve or use the resources that are available on smart factories in relation to traditional factories (there are several strong claims without citations or explanation)?

2. Although there are a limited number of papers on the subject a search for "industrial IOT EKB model" return some positive results. These papers were no added to the sota.

3. The claim on line 92 is too strong for the presented SOTA, how does the Information System Success Model relate to the EKB model? The added text did not make the link between EKB model and System Success Model any clear.

Reviewer 4 Report

I accept the responses and corrections introduced by the authors.

Round 3

Reviewer 2 Report

Most of my concerns were answered.

However, I do not believe that the paper has enough novelty to be considered.

It is trying to create a link between IoT Smart Factories and the proposed model. There is little evidence that IoT would be a factor in the remaining analysis.

Author Response

Thank you very much for taking your time to review this manuscript. The suggestions have enabled us to improve our work. Please find my revision as the attachments.

However, I do not believe that the paper has enough novelty to be considered.

It is trying to create a link between IoT Smart Factories and the proposed model. There is little evidence that IoT would be a factor in the remaining analysis.

We are grateful for the suggestion. we have added some content to explain the novelty approach and the importance of SFS. Please see Line 98-110 & 510-518.

Line 98-110:

Most of past research lack of complete derivation processes in selection procedures, as dimensions are heterogeneous, some focus on certain production process. Therefore, the purpose of this study is to develop a Smart Factory Recommendation System (SFRS) to assist enterprises to select an appropriate SFS by SFS Suppliers Ranking, SFS Suppliers Scoring and SFS Purchase Ability Analysis. It combined Rapid Application Development Method and Modified Delphi Method to derive SFS recommendation indicators and proposed 4-Stage system development methodology including recommendation model construction, prototype indicators construction, amendment & simplification of prototype indicator and recommendation system development. Considering the versatility and regionality of SFS system, the design and selection of dimensions and indicators will target industry’s environment in Taiwan and apply Modified Delphi Method with a large amount of literature review and expert opinions for the scope and limitation in this research.

Line 510-518:

In order to further understand the feasibility and suitability of the system, this study also selected three different manufacturing companies in Taiwan to verify the system ac-cording to different needs. The three manufacturing companies are Qingfa screw manu-facturing, Xiangda plastic manufacturing and Zhanjie steel manufacturing. This research personally reviewed the current status and development needs of the enterprises, and listed the basic functions of smart manufacturing in accordance with the enterprise in de-tail under the recommendation system. At the same time, the supervisor of deci-sion-making and the manufacturing department provided their opinions on IoT that will be used as the basis for further development of the system.

Line 529-537:

The results of the system in this research are for enterprises, they provide solutions and references on purchasing SFS whose functions are (1) to provide purchase ability analysis of SFS; (2) to support demand analysis of manufacture problems; (3) raking and scoring of recommendation indicators and to flexibly delete unnecessary indicators by their de-mands; (4) to automatically calculate recommendation results, which mitigates inconveniences of human forces calculation and provides real-time ranking and the best alter-native recommendation. For SFS Suppliers, results of this study can be a reference on de-sign and modification of the SFS, enabling the SFS's requirements to be closer to users' demands.